# Vitamin D enhances antiviral responses in dengue virus-infected macrophages by modulating early-response gene expression

Yordi Sebastián Tamayo-Molina[1], ☯, Juan Felipe Valdés-López[1], ☯,
Geysson J. Fernandez[2], Silvio Urcuqui-Inchima[1], *

1 Grupo Inmunovirología, Facultad de Medicina, Universidad de Antioquia UdeA, Medellín, Colombia,
2 Grupo Biología y Control de Enfermedades Infecciosas, Universidad de Antioquia UdeA, Medellín, Colombia

☯ These authors contributed equally to this work.
* silvio.urcuqui@udea.edu.co

## Abstract

Dengue virus (DENV), the etiological agent of dengue fever, remains a global health concern, leading to severe illness and death in the absence of any definitive cure. Research has shown that vitamin D may reduce DENV replication *in vitro* and that dengue patients with low or deficient vitamin D levels are at higher risk of severe dengue. Studies indicate that viral replication is inhibited in human monocyte-derived macrophages (MDM) differentiated in the presence of vitamin D (D3MDM), suggesting that vitamin D may prevent DENV entry into host cells. However, despite these findings, the role of vitamin D in regulating the temporal expression patterns of genes as early, mid, and late transcriptional profile of DENV-infected macrophages remains unclear. Therefore, utilizing a kinetic transcriptomic profile is crucial. This approach provides detailed insights into the dynamic changes in gene expression over time, helping to clarify how vitamin D can modulate the immune response at critical stages of DENV infection. To address the transcriptional dynamics, we conducted a comprehensive analysis of gene expression patterns in MDM and D3MDM infected with Dengue virus serotype 2 (DENV-2). Utilizing bulk RNA sequencing alongside a standard viral growth curve, we systematically analyzed transcriptional kinetics by selecting key time points: 1.5, 3, 5.5, and 10 hours post-infection (h.p.i.) to monitor early viral entry and replication events and 24 h.p.i. to assess gene expression during peak viral particle production. Our temporal analysis revealed a progressive increase in cellular transcripts within the first hour of infection, with a more pronounced gene expression pattern in DENV-2-infected MDM compared to DENV-2-infected D3MDM at this early stage. Enrichment analysis indicated a reduced inflammatory response in DENV-2-infected D3MDM. Additionally, transcription factor analysis suggested diminished NF-κB signaling, but enhanced IRF5 activity was elevated in the DENV-2-infected D3MDM. High-dimensional clustering analysis identified nine unique gene

**Data availability statement:** Data was stored in the Gene Expression Omnibus (GEO) repositories under accession number: GSE297386.

**Funding:** SUI; Minciencias, grant grant No. 111584467188, Contract No. 428-2020. https://minciencias.gov.co/sala_de_prensa/se-extiende-la-fecha-para-aplicar-la-convocatoria-icgeb-research-grants-2020-y-se. Universidad de Antioquia-CODI Acta No. 2020-34065. https://www.udea.edu.co/wps/portal/udea/web/inicio/!ut/p/z1/04_Sj9CPykssy0xPLMnMz0vMAfIjo8zi_QJNXQ2NnA18_D2NXQ0CLf1MA4zdPY1MzI31wwkpiAJKG-AAjgZA_VFgJRaWzkaGjiYGPgbehqYGjoGuAX4h_s4BLI5GUAV4zCjIjTDIdFRUBAC3KPbO/dz/d5/L2dBISEvZ0FBIS9nQSEh/. The funders played no role in the study design, data collection and analysis, publication decision, or manuscript preparation.

**Competing interests:** The authors have declared that no competing interests exist.

clusters across both macrophage types, with notable upregulation of genes associated with antiviral activity, including *IDO1, ISG20, OASL, IFI44L, RSAD2, IFIT1, MX1, EPSTI1, CXCL10,* and *CXCL11* in DENV-2-infected D3MDM at 1.5 h.p.i., suggesting an enhanced early antiviral response. These findings indicate that vitamin D modulates the magnitude and diversity of the early transcriptional responses, highlighting its potential as a therapeutic option to mitigate DENV severity.

## Introduction

Dengue virus (DENV) is a vector-borne flavivirus transmitted by the *Aedes* mosquito. According to the World Health Organization (WHO), DENV has been a global public health concern since the 1960s. It remains prevalent in the Caribbean, Latin America, Southeast Asia, and the Pacific Islands [1]. DENV is the etiological agent of dengue fever (DF), or severe forms of the disease known as severe dengue (SD). The physiopathology of SD is multifactorial, but some individuals have a higher risk of developing severe symptoms, such as in cases of heterotypic infection and individuals with co-morbidities, including obesity and diabetes [2–4]. Despite the socioeconomic and public health impact of dengue outbreaks, no antiviral therapy for dengue is currently available. Therefore, further research is urgently needed to develop drugs or treatments to manage DENV infection and prevent the progression of dengue with clinical warning signs.

In the immunopathogenesis of dengue, tissue-resident macrophages are susceptible and permissive to dengue viral replication. Macrophages can detect DENV via pattern recognition receptors (PRRs), including Toll-like receptors (TLRs) and retinoic acid-inducible gene-I (RIG-I)-like receptors (RLRs). Specifically, surface TLR2, along with its co-receptors CD14 and TLR6, has been documented as an innate sensor of DENV [5,6]. In addition, endosomal TLRs such as TLR3, TLR7, TLR8, and cytosolic RLRs, including RIG-I and melanoma differentiation-associated gene 5 (MDA5), recognize DENV viral mRNA [7]. Stimulation of nuclear factor kappa B (NF-κB)- and interferon regulatory factors (IRFs)-dependent gene transcription is a central aspect of inflammatory and antiviral activation by TLRs and RLRs [8]. The NF-κB pathway is essential for generating an inflammatory response in macrophages, including the production and secretion of cytokines, chemokines, and the release of antimicrobial peptides [9]. The IRFs, including IRF1, IRF3, IRF5, and IRF7, have been shown to coordinately regulate type I/II IFNs responses, as well as expression of IFNs stimulated genes (ISG), thereby establishing an antiviral state during DENV infection [10,11]. The establishment of antiviral state, along with the recruitment and activation of inflammatory cells are essential components of the host defense against viral infections. However, DENV can subvert these antiviral pathways, delaying the host's defense mechanism and amplifying viral replication, which may lead to an excessively vigorous response that can be detrimental, potentially causing plasma leakage, organ dysfunction and increasing the risk of mortality [12]. Clinical data have shown

an association between levels of CC chemokines in plasma and the severity of dengue [13,14]. Further, inflammatory chemokines have indeed been implicated as a contributor to the pathogenesis and severity of DENV infection [15–17]. Therefore, modulating DENV-induced chemokines may potentially prevent excessive inflammation and the induction of the so-called cytokine storm [18–20].

In this context, 1,25-dihydroxyvitamin $D_3$ (1,25(OH)$_2$D$_3$), hereafter referred to as Vitamin D, has emerged as a prophylactic and a good treatment against viral infections, including DENV. Recent studies have identified a link between low vitamin D levels and heightened infection susceptibility [21]. Furthermore, it has been reported that daily supplementation with 4000 international units of Vitamin D decreased the susceptibility of monocyte-derived dendritic cells (MDDC) and monocyte-derived macrophages (MDM) to DENV-2 infection [22,23]. In addition, MDM differentiated in the presence of vitamin D (D3MDM) exhibit reduced surface expression of macrophage mannose receptor (MMR/CD206), thereby restricting DENV entry [24]. Moreover, D3MDM show higher expression of antiviral proteins and antimicrobial peptides, which interfere with DENV-2 [25,26] and Zika virus replication [27]. RNA-Seq and miRNA-Seq analysis of D3MDM revealed that Vitamin D enhances the immune response of macrophages by modulating a regulatory network of microRNAs and mRNAs, which plays a critical role in orchestrating the transcriptional reprogramming necessary to combat viral infections [28]. These antiviral effects fine-tune the immune response, ensuring that macrophages can efficiently control viral replication while avoiding excessive inflammation that could lead to severe outcomes.

The experiments using transcriptional kinetics of macrophages in response to lipopolysaccharide exposure or coronavirus infection have revealed significant early, mid, and late gene responses, underscoring the dynamic regulation of transcriptional profiles [29,30]. On the other hand, vitamin D target genes, in a time course analysis in mononuclear cells, are classified into primary target genes (4–8 h) and secondary targets (24–48 h) [31]. Accordingly, studying the transcriptional kinetics provides a framework to investigate how the temporal transcriptional response of vitamin D target genes, segregated into early, mid, and late targets, may not only mitigate inflammation but also enhance the dynamics of the macrophage responses against dengue virus.

Given the predominant role of Vitamin D in limiting virus replication, this study aims to utilize bulk RNA-Seq to analyze MDM and D3MDM infected with DENV-2. To achieve this, we used a standard viral growth curve to examine the early, mid, and late stages of infection in MDM and D3MDM. Our results show that in D3MDM, predicted early IRF5 transcription activity leads to the expression of a subset of ISGs that can restrict viral replication, which results in reduced early expression of genes involved in the pro-inflammatory response. We propose that, in D3MDM, vitamin D restricts early viral replication, thereby dampening the initial inflammatory response.

## Materials and methods

### Ethics statement

The study was approved by the Ethics Committee of the "Sede de Investigación Universitaria-Universidad de Antioquia." Written informed consent was obtained from all individuals who voluntarily participated in this study according to the principles of the Declaration of Helsinki. Four healthy donors were included in this study.

### DENV-2 stock and viral titration

DENV-2 stock was obtained by growing the virus in *Aedes albopictus*-derived C6/36-HT cells (ATCC) at a multiplicity of infection (MOI) of 0.01. C6/36 HT cells were grown in Leibovitz's L-15 medium (L-15; Sigma-Aldrich) supplemented with 5% heat-inactivated fetal bovine serum (FBS; Gibco, Thermo Fisher Scientific, Massachusetts, USA) and 1% antibiotic-antimycotic solution (Corning, New York, USA), and incubated at 34 °C. DENV-2 culture supernatants were harvested, stored at −80 °C, and titrated by plaque assay on BHK-21 cells (clone 15, ATCC).

## Culture of primary human monocytes and differentiation into monocyte-derived macrophages in the presence (D3MDM) or absence (MDM) of vitamin D

Human peripheral blood mononuclear cells (PBMCs) from leukocyte-enriched blood units of healthy donors were isolated through a density gradient with Lymphoprep (STEMCELL Technologies Inc., Vancouver, Canada) by centrifugation at 850 x g for 21 min. Platelet depletion was performed by washing with PBS 1X (Sigma-Aldrich) three times at 250 x g for 10 min, and the percentage of CD14-positive cells was determined using flow cytometry. To obtain human monocytes, 24-well plastic plates were scratched with a 1000 µL pipette tip and seeded with $5x10^5$ CD14 positive cells per well to allow its adherence during 2 h in RPMI-1640 medium (Sigma-Aldrich) supplemented with 0.5% autologous serum, 4 mM L-glutamine, and 0.3% NaCO3 and incubated at 37°C and 5% $CO_2$. Non-adherent cells were removed by washing twice with 1X PBS. Monocytes were cultured in RPMI-1640 medium supplemented with 10% FBS, 4 mM L-glutamine, 0.3% NaCO3, and 1% antibiotic-antimycotic solution (100X), either with (D3MDM) or without (MDM) 0.1 nM or 1 nM of vitamin D. The cells were incubated at 37°C and 5% $CO_2$ for 6 days to differentiate into MDMs, with fresh media added to the cultures every 2 days (S1 Fig).

## In vitro DENV-2 infection of monocyte-derived macrophages

MDM and D3MDM were infected with DENV-2 strain from New Guinea at an MOI of 5 in serum-free RPMI-1640. The samples were incubated at 37 °C for 1.5 hours, after which the cells were washed with 1X PBS to remove unbound virus, and a fresh complete medium was added. The cells were then incubated at 37 °C and 5% $CO_2$. Culture supernatants and cell lysates were collected at 1.5-, 3-, 5.5-, 10-, and 24-hour post-infection (h.p.i.) and stored at −80 °C (see S1 Fig).

## RNA extraction and cDNA synthesis

Total RNA was extracted using the Direct-zol™ RNA Miniprep Plus kit (Zymo Research, California, USA), following the manufacturer's protocol. RNA samples were treated with a DNase I column (Zymo Research) to remove contaminating genomic DNA. RNA concentration was measured using Nanodrop spectrophotometry (Thermo Scientific, Massachusetts, USA). For bulk RNA-Seq, 1 µg of RNA was used per sample, and sequencing was performed on an Illumina HiSeq 2000 platform (Macrogen, Seoul, South Korea). After sequencing, the image data were converted into raw reads and stored in FASTQ format for each sample, with quality assessment using FastQC [6].

## Bulk RNA-Seq of DENV-infected macrophages

Clean reads were obtained by removing low-quality adapter, poly N-containing, and shorter-than-70 bp reads. The location of the reads on the reference genome was determined quickly and precisely by comparing reads with the reference genome (GRCh38 or dengue virus type 2 [NC_001474]) using HISAT2 software [7]. The new transcripts were then assembled using StringTie software [8], and using the feature Counts tool in Subread software [9], the raw count number in each sample was obtained. Data was stored in the Gene Expression Omnibus (GEO) repositories under accession number: GSE297386.

## Data annotation and batch effect correction

The raw counts were processed using the following workflow in R software (version 4.2.0) [10]. First, gene rows were annotated with their respective genotype (non-coding gene, pseudogene, and protein-coding gene), symbols, and Entrez IDs. We then filtered the dataset to include only protein-coding genes and generated a list of their lengths using the *Homo.sapiens* library [11]. Batch effect correction of the raw count data for each transcriptome was performed using ComBat-Seq [12].

## Analysis of differentially expressed genes

To identify the top differentially expressed genes (DEGs), we selected genes with a false discovery rate (FDR) < 0.05 and |Log$_2$ Fold Change (FC) (treated/untreated) > 1. Multidimensional scaling (MDS) analysis plot was performed using Glimma [14], and Pearson correlation between samples was calculated (see S2A Fig). Gene set enrichment analysis (GSEA) for overall genes regulated in MDM and D3MDM were performed using the ClusterProfiler package (Version 4.14) [15], while transcription factor analysis was performed with the TFactS database (Version 0.99) [16] (see S2B Fig). Unique DEGs between the two transcriptomes were identified, followed by GSEA using ClusterProfiler (Version 4.14) [15] and transcription factor analysis with iRegulon (Version 1.3) [17] (see S2C Fig).

## Statistical analysis

Statistical analyses and figure generation were performed using R statistical software. Anova was used to compare treatments between the groups, and for multiple comparisons, Student's T-test with Bonferroni correction was applied. For RNA-seq statistics, we use DESeq2 library [13], and to enhance data visualization, we utilized ggplot2 (Version 3.5) [32], ggprism (Version 1.0) [33], and ComplexHeatmap (Version 2.22) [34].

## Results

### The expression of genes in the VDR signaling pathway and their target genes depend on vitamin D concentration

Macrophages are key target cells for DENV infection and are crucial in orchestrating the innate immune response to control viral replication. In this study, we performed a comparative analysis of macrophages differentiated in the presence of physiological doses of vitamin D (D3MDM) or not (MDM) and infected with DENV-2 serotype 2 New Guinea strain (MOI: 5). The concentration of vitamin D used in this study (0.1 nM) was selected to mimic physiological levels of circulating 1,25(OH)$_2$D$_3$, which typically range between 0.04 and 0.150 nM in serum. This concentration also corresponds to the high-affinity dissociation constant (Kd = ~0.1 nM) of 1,25(OH)$_2$D$_3$ for the VDR [35]. Furthermore, we previously reported that macrophages differentiated in the presence of 0.1 nM vitamin D (D3MDM) exhibited reduced susceptibility to DENV infection [24,25]. As shown in Fig 1A, D3MDM were consistently less susceptible to DENV-2 infection than MDM. The percentage of inhibition reached approximately 35% and 45% in D3MDM differentiated with 0.1 nM (D3[0.1 nM]MDM) and 1 nM (D3[1 nM]MDM) of vitamin D, respectively, compared to MDM at 24 h.p.i. (Fig 1B). It is worth noting that the antiviral effect of vitamin D appears to be dose dependent.

To further gain a better understanding of the effect on the vitamin D signaling pathway, we analyzed the VDR signaling pathway using our bulk RNA-Seq datasets of MDM differentiated with a physiological dose of vitamin D (D3[0.1 nM]MDM), and compared them to publicly available transcriptomic data from MDM differentiated with 1 nM vitamin D ((D3[1 nM] MDM; GSE209698). The DEGs analysis revealed 16 upregulated and 43 downregulated genes in D3[0.1 nM]MDM, while 100 genes were upregulated and 56 downregulated in D3[1 nM]MDM (Fig 1C). Gene overrepresentation analysis using WikiPathways showed that both concentrations induced the expression of genes associated with the vitamin D receptor pathway (WP2877), including *CLMN*, *G0S2*, *CD14*, *TREM1*, and *CAMP*, which were regulated in both concentrations (Fig 1D). In contrast, the higher vitamin D concentration also induced the expression of additional genes, such as *CYP24A1*, *MYC*, *ALOX5*, *HILPDA*, and *S100A8* (Fig 1E). These findings indicate that 0.1 nM of vitamin D is insufficient to drive notable transcriptional changes, whereas 1 nM exerts a more pronounced effect on VDR-regulated genes.

### Analysis of differentially expressed genes during early and late stages of DENV-2 infection in macrophages differentiated in the presence of vitamin D

To obtain a broad picture of the MDM and D3MDM response to DENV-2 infection, bulk RNA-Seq was used to analyze differential gene expression at the mRNA level. The RNA was isolated at 1.5-, 3-, 5.5-, 10- and 24- h.p.i. and converted

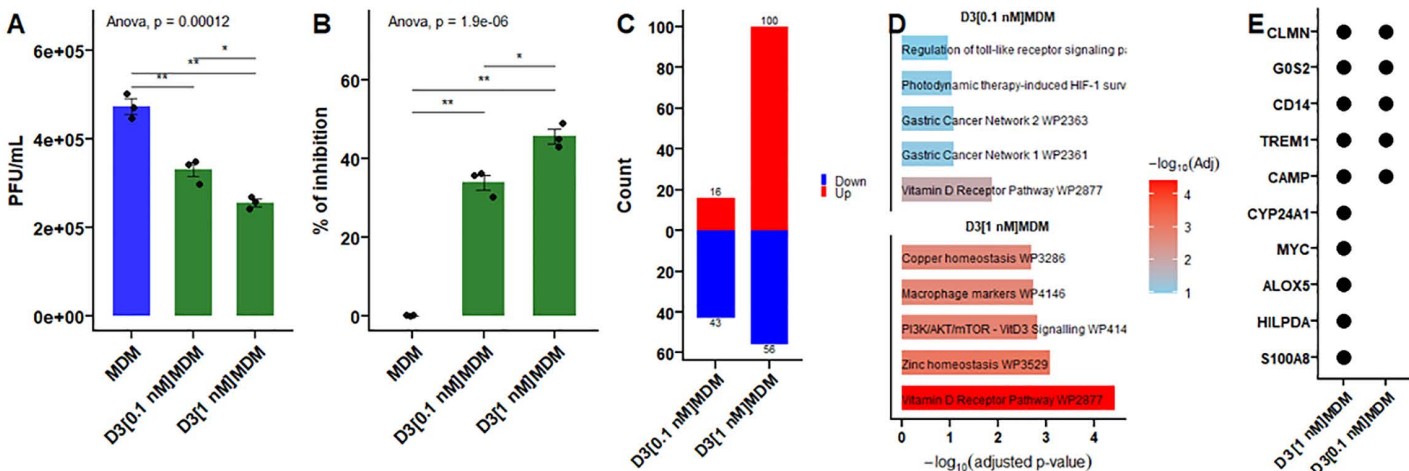

**Fig 1. Virus quantification and VDR pathway analysis. (A)** Plaque assay of MDMs differentiated in the presence of 0.1 nM (D3[0.1 nM]MDM) or 1 nM (D3[1 nM]MDM) of vitamin D and infected with DENV-2 at 24 h.p.i. **(B)** Percentage of viral inhibition in D3[0.1 nM]MDM and D3[1 nM]MDM compared to MDMs. **(C)** Number of differentially expressed genes in uninfected D3[0.1 nM]MDM and D3[1 nM]MDM. **(D)** Overrepresentation analysis of enriched signaling pathways. **(E)** Common and unique VDR-regulated genes in D3[0.1 nM]MDM and D3[1 nM]MDM. Data are presented as mean ± standard deviation.

into cDNA libraries for sequencing. Two biological replicates were processed for each condition. The early time points (1.5-, 3-, and 5.5- h.p.i), and the mid time point (10 h.p.i). were selected to capture kinetic changes in the transcriptional profile associated with early viral entry and replication, independent of the effects of infectious virus release. The 24 h.p.i. time point was chosen as the late time of infection, corresponding to the assembly and release of new viral progeny. Gene counts of viral mRNA indicated a delay in viral replication at the mid time point (10 h.p.i.) in D3MDM compared to MDM (Fig 2A). However, a slightly higher count of viral mRNA was observed at earlier time points (1.5-, 3-, and 5.5- h.p.i) in D3MDM compared to MDM. Interestingly, there were no differences in the count of viral mRNA between the two macrophage types at 24 h.p.i. (Fig 2A). This suggests that vitamin D3 may have a differential effect on viral replication kinetics during the early- and mid-stages of DENV-2 infection.

Next, we investigated the time course of macrophage transcriptomic responses to DENV-2 infection. To analyze the transcriptome data, the transcripts corresponding to protein-coding genes were subjected to principal component analysis (PCA) to identify changes in gene signatures (see S3A Fig). A PCA plot generated from differentially expressed genes (DEGs) revealed that at early- and mid-stages of infection (1.5-, 3-, 5.5-, and 10- h.p.i.), the principal components (PC) 1 and 2 accounted for 53% and 25% of the total variance, respectively (Fig 2B). However, a major transcriptional shift occurred at 24 h.p.i, reflected by a dramatic change in the variance; PC1 explained 94% of the variance, while PC2 explained only 6% (Fig 2C).

To define the top DEGs, we selected genes with a p-adj < 0.05 and |$\log_2$ FC (Infected MDM or D3MDM/Uninfected MDM or D3MDM) | > 1. Results show that MDM exhibited more DEGs than D3MDM at all times of infection evaluated, except at 24 h.p.i. A total of 2.941 genes exhibiting differential expression were identified following DENV-2 infection of MDM, with 1.953 upregulated and 988 downregulated genes. Specifically, at 1.5-, 3-, 5.5-, 10-, and 24- h.p.i, there were 82, 255, 215, 149, and 1.252 upregulated genes, while 3, 45, 122, 108, and 710 were downregulated genes, respectively (Fig 2D). In the same way, a total of 2.140 DEGs were identified following DENV-2 infection of D3MDM, with 1.553 upregulated and 584 downregulated. At 1.5-, 3-, 5.5-, 10-, and 24- h.p.i, 63, 132, 89, 29, and 1.240 genes were upregulated, while 5, 13, 6, 0, and 560 were downregulated, respectively (Fig 2D). Moreover, based on the distinct

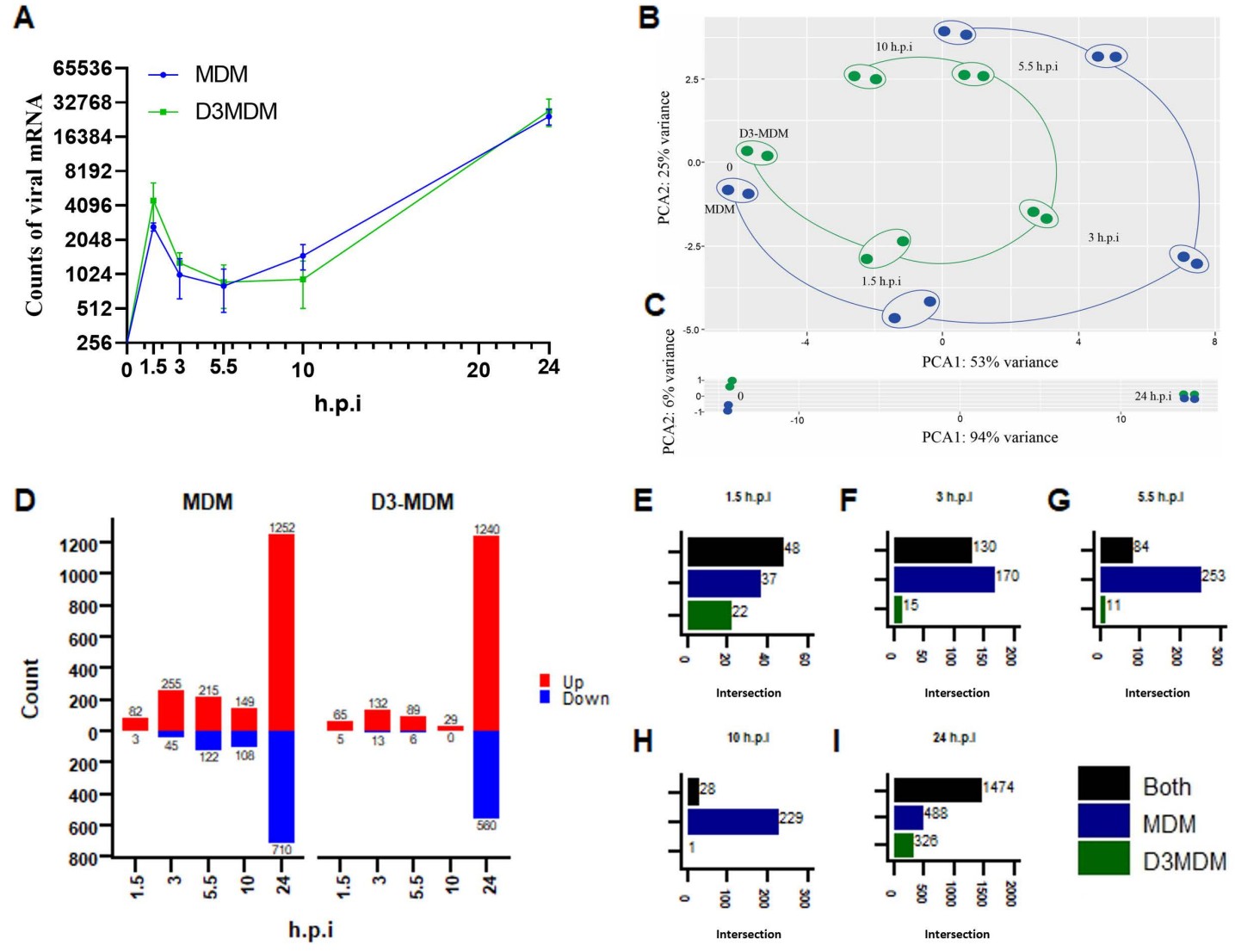

**Fig 2. Early and late response of MDM and D3MDM to DENV-2 infection. (A)** Kinetics of DENV-2 genome copies over time. **(B)** Principal component analysis (PCA) showing the transcriptional trajectories of MDM (blue) and D3MDM (green) at 0-, 1.5-, 3-, 5.5-, and 10-hpi. **(C)** PCA for the early (0 h) and late (24 h) time points. **(D)** Number of upregulated and downregulated genes over time in MDMs and D3MDMs. **(E-I)** Shared and unique differentially expressed genes at 1.5- **(E)**, 3- **(F)**, 5.5- **(G)**, 10- **(H)**, and 24-hpi **(I)**. Data are presented as mean ± standard deviation.

transcriptional trajectories observed in the PCA, correlation analysis between MDM and D3MDM suggests that early infection DEGs are highly correlated and time-dependent (see S3B Fig).

We compared the number of DEGs between macrophages to screen for a common or distinct transcriptional signature. Both macrophage types (MDM and D3MDM) exhibit a shared core transcriptional response following DENV-2 infection, regulating 48 (Fig 2E), 130 (Fig 2F), 84 (Fig 2G), 28 (Fig 2H), and 1.474 (Fig 2I) common genes at 1.5-, 3-, 5.5-, 10-, and 24- h.p.i, respectively. Furthermore, DENV-2-infected MDMs exhibited a more diverse response, with 39, 166, 252, 228, and 487 uniquely expressed genes compared to 22, 15, 11, 1, and 327 genes in D3MDM at the times of infection evaluated (Fig 2E-I). Overall, these results showed a higher gene expression pattern in DENV-2-infected MDM than in DENV-2-infected D3MDM at the early- and mid-stages of infection, with a delayed DENV-2 viral mRNA replication.

## Vitamin D shapes the gene expression in DENV-2-infected macrophages

Gene set enrichment analysis (GSEA) using clusterProfiler revealed distinct transcriptional signatures in DENV-2-infected MDM and D3MDM (Fig 3A). GSEA of DEGs found in DENV-2-infected macrophages identified 32 enriched active GO functions. However, the set of genes associated with these GO terms showed significant differences between the two groups. In both macrophage types, across different post-infection time points, the main activated pathways involved signal transduction, inflammatory response, response to cytokines, and cytokine-mediated signaling pathway. Additional enriched pathways in DENV-2-infected MDM included response to chemokine, chemokine-mediated signaling pathway, leukocyte chemotaxis, granulocyte chemotaxis, and neutrophil chemotaxis. In contrast, these pathways were less enriched in DENV-2-infected D3MDM, particularly during the mid-stage of infection (10 h.p.i.).

Interestingly, specific pathways related to toll-like receptors signaling, interleukin (IL) 1 beta production, macrophages chemotaxis, and natural killer cell chemotaxis were enriched at various post-infection time points in DENV-2-infected MDM but not in DENV-2-infected D3MDM (Fig 3A). Conversely, the regulation of cytoskeletal organization was uniquely enriched in DENV-2-infected D3MDM. Pathways such as IL-10 production, IL-12 production, and T-cell chemotaxis were enriched in MDMs and D3MDM, but only at 24 h.p.i (Fig 3A). A notable distinction was observed in the viral response pathway. In DENV-2-infected MDMs, this pathway was enriched at multiple time points, with the most substantial enrichment at late stage (24 h.p.i.), but no enrichment at mid stage (10 h.p.i.). In contrast, DENV-2-infected D3MDM displayed enrichment of the viral response pathway only at early (1.5- and 5.5- h.p.i.) and late (24- h.p.i.) time points. Similarly, the stress response pathway was consistently enriched from the earliest time points in DENV-2-infected MDM, while in D3MDM, this pathway was enriched at three distinct peaks: at 1.5-, 5.5- and 24- h.p.i., (Fig 3A). In general, the pattern of GO functions enriched at early- and mid-time post-infection were less similar between MDM and D3MDM than the pattern observed at late stage.

In summary, while the enriched GO functions at 24 h.p.i. were broadly similar between MDM and D3MDM, significant differences were evident at earlier and mid time points, particularly at the mid stage (10 h.p.i.). This suggests that D3MDM may exhibit a unique early transcriptional response to DENV infection, which might influence the subsequent inflammatory and antiviral responses.

## Global changes in gene expression may be linked to the activity of NF-κB

The distinct transcriptional patterns observed in DENV-2-infected MDM and D3MDM emphasized the dynamic and complex nature of the immune response. Global changes in gene expression, such as those linked with the delayed inflammatory response of DENV-2-infected D3MDM, may be associated with the differential accumulation of transcription factors (TFs). To address this, we used TFactS in R software to identify key TFs-driven gene expression differences between MDM and D3MDM infected with DENV-2.

The top 14 most influential TFs common to both infected macrophage types and that was part of the core enrichment group included *AR*, *ATF3*, *BACH1*, *CEBPB*, *EGR1*, *JUN*, *NFKB1*, *RELA*, *SMAD3*, *SMAD4*, *TP63,* and *VDR* (Fig 3B). Although these TFs were regulated from the early hours of infection, their activity varied significantly over time, particularly during the early- and mid-stages of infection. The most pronounced differences occurred at the mid stage (10 h.p.i.). However, at 24 h.p.i., both macrophage types exhibited a similar activity of TFs, suggesting a convergent response during the late stage of infection. Notably, the number of genes regulated by TFs associated with inflammatory responses (*RELA*, *NFKB1,* and *JUN*) and with vitamin D signaling pathway (*VDR*), peaked at 3-, 5.5- and 24- h.p.i. in DENV-2-MDM, whereas in DENV-2-infected D3MDM, peaks were observed only at 3- and 24-h.p.i. (Fig 3C). In addition, we observed temporal changes in the expression of these transcription factors during infection. In both macrophage types, *VDR* mRNA levels declined from 0 to 3 h.p.i., peaked at 5.5 h.p.i., and then decreased again from 10 to 24 h.p.i. (Fig 3D) Notably, *VDR* expression was consistently higher in D3MDMs than in MDMs between 0 and 3 h.p.i. We also found that *JUN*, *RELA*,

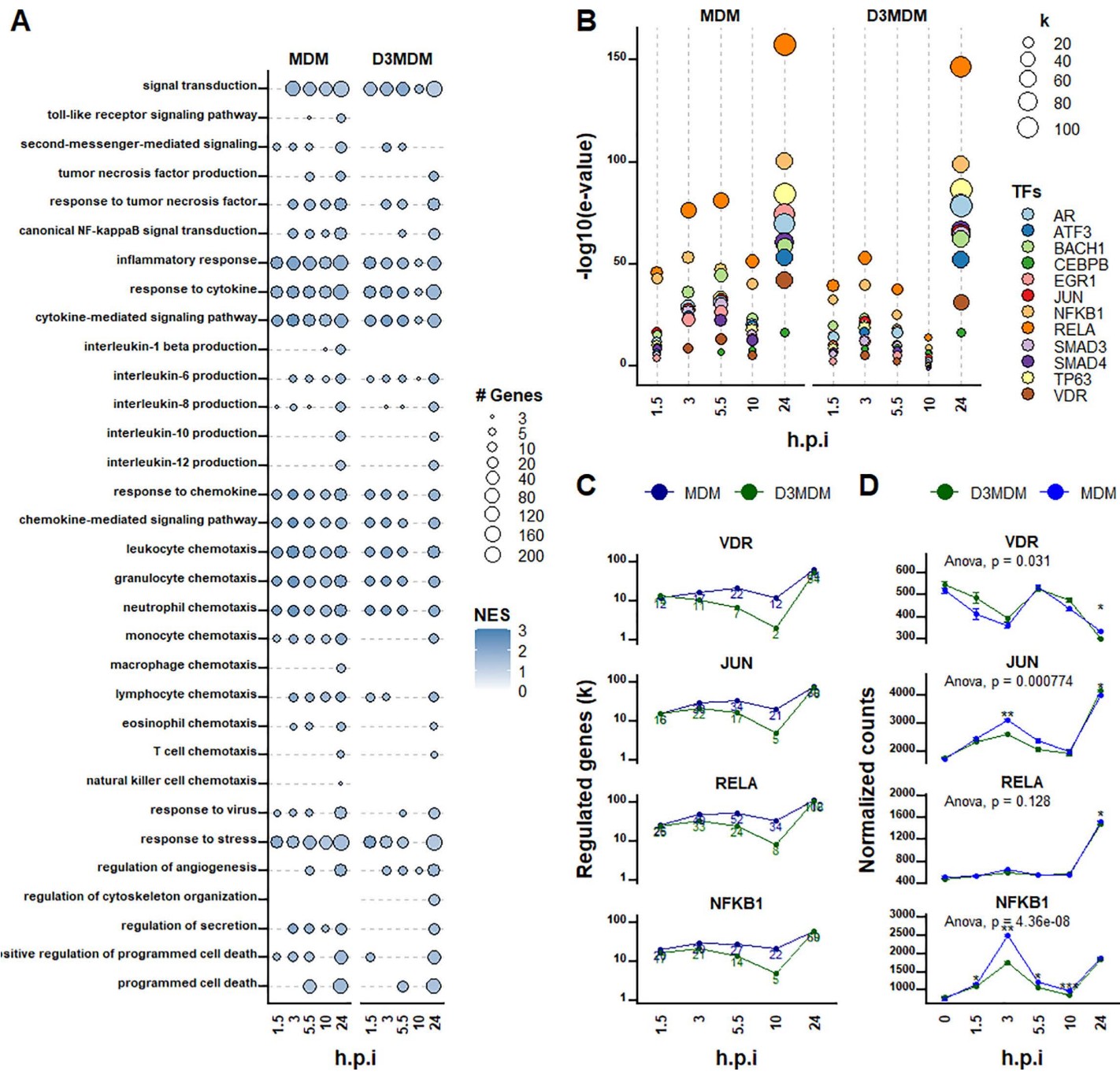

**Fig 3. Vitamin D modulates gene expression in DENV-2-infected MDM and D3MDM. (A)** Gene set enrichment (GSE) analysis of DEGs between MDMs and D3MDMs. **(B)** Transcription factor (TF) enrichment analysis of GSE-identified genes using TFactSR. **(C)** The TFs VDR, JUN, RELA, and NFKB1 regulated the number of target genes over time. **(D)** Normalized expression counts of the genes encoding these TFs at different time points post-infection. Data are presented as mean ± 95% confidence interval.

and *NFKB1* mRNA expression peaked at 3 h.p.i., declined at 10 h.p.i., and increased again at 24 h.p.i. Also, at their peak expression, the levels of these TFs were lower in D3MDMs than in MDMs, with the most pronounced difference observed for *NFKB1* (Fig 3D). This cyclical pattern of NF-κB-dependent inflammatory gene expression suggests a more robust and sustained inflammatory response in MDMs during the early- and mid-stages of infection compared to D3MDM.

## Unique genes are dependent on the transcription factors NF-κB and IRF5

The results presented so far show that differentiated macrophages' gene expression levels and patterns differ in the presence or absence of vitamin D at early- and mid-stages post-infection. To investigate the mutually exclusive effect of vitamin D on gene expression, we focused on unique genes expressed in DENV-2-infected MDM and DENV-2-infected D3MDM (Fig 2E-I). We generated a network of TFs and their target genes using iRegulon to determine the relationship between unique genes highly enriched in the two groups of macrophages. We found that *NFKB1* is predicted to be a TF-regulating gene expression in DENV-2-infected MDM, while both *IRF5* and *NFKB1* are expected to regulate the expression pattern in DENV-2-infected D3MDM, each with a normalized enrichment score (NES) > 6 (Fig 4A). The predicted motifs are shown in the inset of Fig 4A.

As indicated in the network, the NF-κB complex was enriched for regulons in both DENV-2-infected MDM and D3MDM, whereas IRF5 was exclusively predicted in D3MDM (Fig 4B). To determine whether vitamin D directly regulates *IRF5* and *NFKB1* gene expression, we reanalyzed VDR ChIP-Seq data (GSE89431) and transcriptomic data from wild-type (TW) and VDR-knockout (KO) THP-1 cells (GSE157514), at both early (2 and 4 h) and late (24 h) time points. VDR binding was consistently observed at the proximal promoter of *IRF5* from early to late time points, whereas no VDR occupancy was detected at the *NFKB1* locus (Fig 4C). Accordingly, *IRF5* transcript levels (TPM) increased significantly in vitamin D-treated WT THP-1 cell (D3-VDR-THP1), but not in vitamin D-treated VDR-KO THP-1 cells (D3-KO-THP1), where *IRF5* expression remained comparable to that of untreated controls (Fig 4D). This demonstrates that the gene expression differences at early stages between both macrophage types, whether differentiated in the presence or absence of vitamin D, are *IRF5* dependent.

## The time course of NF-κB and IRF5 target gene expression in MDM and D3MDM

We performed hierarchical clustering analysis to identify unique gene expression patterns in DENV-2-infected MDM or D3MDM, revealing nine distinct clusters across the viral growth curve (Fig 5).

Cluster 1 (n = 21 genes) displayed higher gene expression levels in DENV-2-infected MDM than in D3MDM (Fig 5A). These genes peaked in expression at 3 h.p.i. in both macrophage types, followed by a decline at 10 h.p.i. and then reached a second peak at 24 h.p.i. Notably, at 10 h.p.i, gene expression was almost completely abolished in DENV-2-infected D3MDM but recovered to levels comparable to MDM by 24 h.p.i. Genes in this cluster were mainly associated with cytokine-mediated signaling pathways, response to tumor necrosis factor, and leukocyte chemotaxis.

Cluster 2 (n = 20 genes) comprised genes exclusively transcribed in DENV-2-infected MDM at early time points, with a peak expression at 3 h.p.i. followed by near-complete suppression at 10 h.p.i. (Fig 5B). These results suggest that vitamin D treatment may downregulate the expression of these genes. Cluster 2 genes were mainly involved in cytokine-mediated signaling pathways and leukocyte chemotaxis.

Cluster 3 (n = 11 genes) comprised genes whose maximum expression peak was observed in DENV-2-infected MDM at 3 h.p.i. whereas these genes were not expressed in D3MDM at this time point (Fig 5C). Notably, the expression of genes of this cluster was absent at 5.5- and 10- h.p.i. in both macrophage types. However, by 24 h.p.i. the expression became similar in both MDM and D3MDM. This cluster includes genes associated with cytokine-mediated signaling pathways, IL-8 production, and positive regulation of programmed cell death.

Cluster 4 (n = 24 genes) included genes with low expression levels at the early infection stages, specifically in DENV-2-infected MDM (Fig 5D). However, both macrophage types displayed a maximum expression peak of these genes at

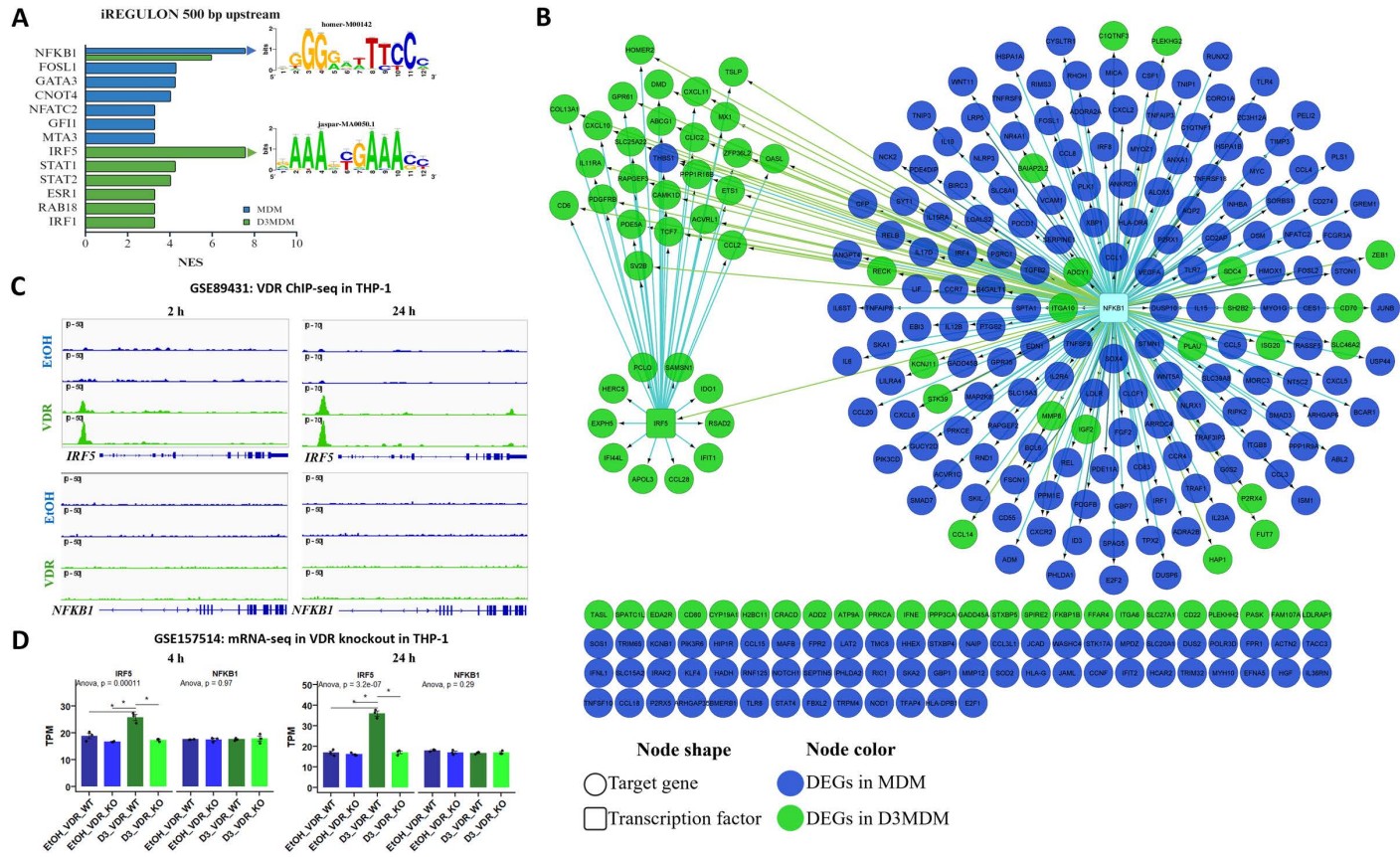

**Fig 4. Analysis of TF target genes unique to MDM and D3MDM during DENV-2 infection.** (A) Enrichment of TFs and associated gene motifs predicted by iRegulon for *NFKB1* and *IRF5*. (B) The regulatory network of TFs and their regulons, highlighting unique target genes in MDM and D3MDM. (C) VDR ChIP-seq analysis in THP-1 cells at early (2 h) and late (24 h) time points showing VDR binding near *IRF5* and *NFKB1* gene loci. (D) mRNA-seq analysis of VDR knockout (VDR-KO) THP-1 cells showing transcript per million (TPM) levels of *IRF5* and *NFKB1* at 4 and 24 h. Data are presented as mean ± 95% confidence interval.

24 h.p.i. The genes of this cluster included genes involved in cytokine-mediated signaling pathways and positive regulation of programmed cell death.

Cluster 5 (n = 27 genes) and cluster 6 (n = 26 genes) included genes induced during the late stage of infection but only in DENV-2-infected MDM. In cluster 5, genes peaked in expression at 24 h.p.i. (Fig 5E), while in cluster 6, gene expression decreased at 24 h.p.i. (Fig 5F). Upregulated genes in these clusters were primarily involved in toll-like receptor signaling pathways and canonical NF-κB signal transduction. In contrast, downregulated genes were mainly associated with cytoskeleton organization and regulation of secretion.

Cluster 7 (n = 13 genes) consists of genes initially induced in DENV-2-infected D3MDM at 3 h.p.i. but not in MDM. These genes were not expressed at 5.5 h.p.i. and reached their maximum expression peak at 24 h.p.i. in both macrophage types (Fig 5G). Genes in this cluster were involved in cytokine-mediated signaling pathway and virus response, including *IDO1*, *ISG20*, *OASL*, *IFI44L*, *RSAD2*, *IFIT1*, *MX1*, *EPSTI1*, *CXCL10*, and *CXCL11* targets of *IRF5* (Fig 4C). These findings suggest that D3MDM upregulates antiviral genes early in the infection, which may influence viral entry, reducing the number of infected cells, as indicated in our previous studies [24,25].

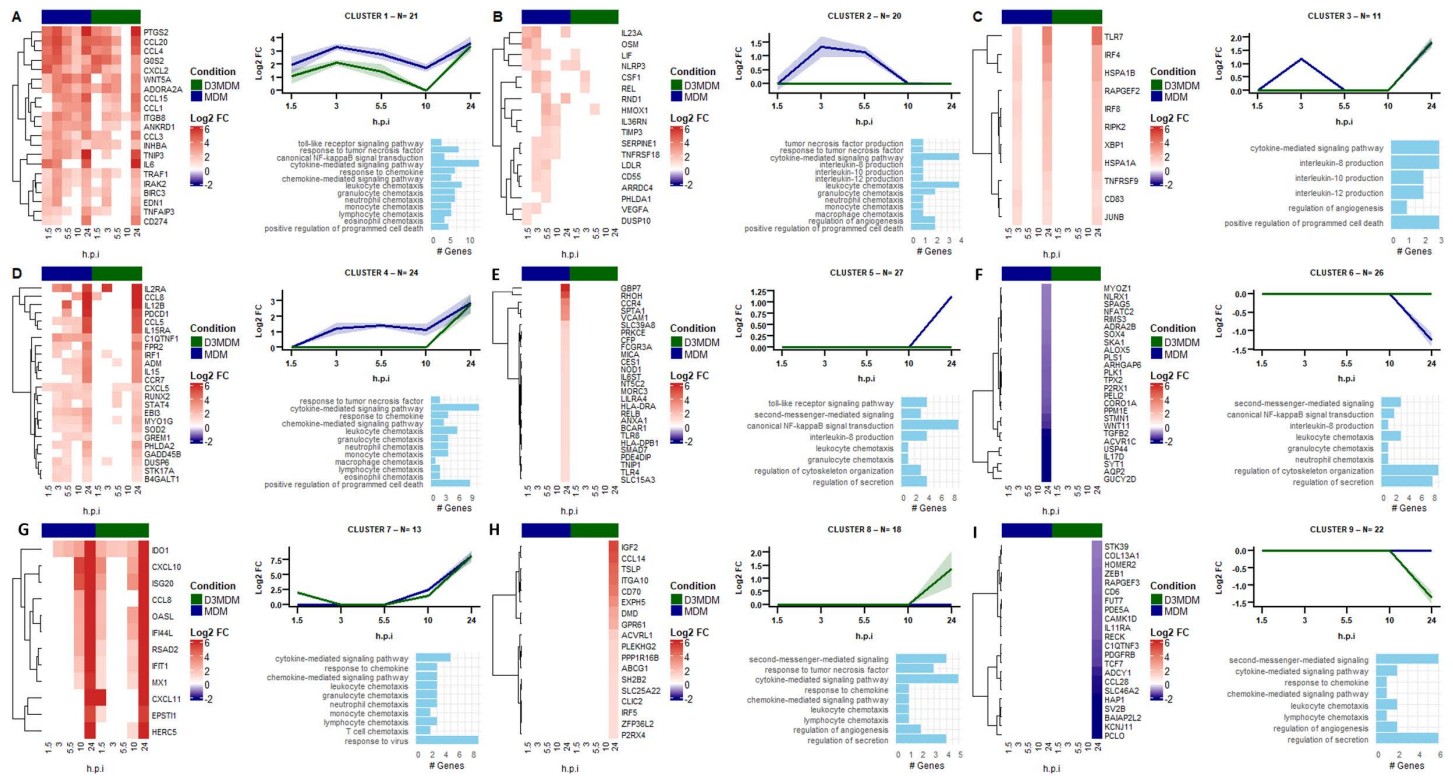

**Fig 5. Hierarchical clustering analysis of unique genes in DENV-2-infected MDM and D3MDM.** Each cluster shows DEGs uniquely regulated at least one time point post-infection in either macrophage type. For each cluster, a heatmap displays the log2FC in gene expression over time; line plots show the temporal expression pattern for MDM (blue) and D3MDM (green), with lines representing the mean ± 95% confidence interval; and a bar plot illustrating the enriched biological processes associated with each gene cluster. **(A)** cluster 1, **(B)** cluster 2, **(C)** cluster 3, **(D)** cluster 4, **(E)** cluster 5, **(F)** cluster 6, **(G)** cluster 7, **(H)** cluster 8, and **(I)** cluster 9.

Interestingly, in contrast to the patterns observed for clusters 5 and 6, cluster 8 (n = 27 genes) and cluster 9 (n = 26 genes) involved genes that were either upregulated or downregulated, respectively, in DENV-2-infected D3MDM starting at 10 h.p.i. and reaching their maximum expression peak at 24 h.p.i. (Fig 5H and I). The upregulated genes in cluster 8 were primarily involved in second messenger-mediated signaling, response to tumor necrosis factor, cytokine-mediated signaling pathway, and regulation of secretion. Conversely, the downregulated genes in cluster 9 were mainly associated with second messenger-mediated signaling and regulation of secretion.

## Discussion

Immunomodulatory compounds have gained attention in recent years and are considered promising therapeutic options for preventing severe dengue fever. In particular, vitamin D has gained interest in this field due to its ability to modulate inflammatory responses [28,36–38]. In the present study, to further explore the mechanism of action of vitamin D in human macrophages, we differentiated primary human macrophages in the presence of physiological doses of Vitamin D (0.1 nM). Using mRNA-Seq analysis, we identified the time-course effect of vitamin D on DENV-2 replication and macrophage transcriptome. As was observed in previous studies conducted in our laboratory [24,25], we found that DENV-2-infected D3MDM exhibited a lower percentage of infected cells (~25%) compared to conventional MDMs (~15%) at MOI of 5. This reduction infection was associated with decreased mRNA expression of *TLR3* and *TLR7*, along with increased expression of antiviral genes, including *PKR* and *OAS1* [22–25]. Here we found in DENV-2-infected D3MDM that kinetics

of mRNA levels of both chemokines and cytokines decreased at early post-infection. However, viral mRNA was higher in the presence of vitamin D at 1.5-, 3- and 5.5- h.p.i. and significantly decreased at 10 h.p.i. In addition, at later stages of infection, DENV-2 viral load decreased in D3MDM, whereas the transcriptional program and viral genome copies remain relatively similar between the two macrophage types. This may indicate that vitamin D dampens the early- and mid-inflammatory response to DENV-2 and delays viral genome replication, promoting a less inflammatory phenotype in macrophages and significantly reducing the expression of genes associated with the recruitment of inflammatory cells. Further, the shared late transcriptional response can be explained by the presence of a bystander-like phenotype [39,40].

Vitamin D is well-documented to exert genomic effects via the transcription factor activity of the VDR on target gene promoters [41]. We observed that macrophages differentiated in the presence of physiological doses of vitamin D have a minor influence on DEGs and VDR target genes. In contrast, at higher concentrations (1 nM), vitamin D upregulates more DEGs and VDR target genes, suggesting that the effects of physiological doses of vitamin D were predominantly genomic. Moreover, according to the literature, previous studies have reported non-genomic effects of vitamin D that alter the expression of genes involved in various signaling cascades, including those related to innate and adaptive immunity [25,42]. However, we could not assess these effects in the present study. On the other hand, we have reported that genomic effects of vitamin D upregulates antimicrobial peptides such LL-37 (encoded by *CAMP*) that restrict Zika virus (ZIKV) and DENV-2 infection in human macrophages [26,27]. However, ZIKV infection has been shown to inhibit the expression of VDR target genes in human monocytes, thereby preventing Vitamin D antiviral effects [43].

The altered transcriptional program observed in D3MDM was driven by the lower activity and expression of TFs-encoding genes such as *NFKB1* and *RELA*, which is in accordance with the downregulation of genes involved in cytokine- and chemokine-mediated signaling pathways. In line with these results, several studies have found that vitamin D decreases DNA binding of NF-κB by increasing the expression of IκBα in keratinocytes [44], fibroblasts [45,46], dendritic cells [47], and kidney cells [48]. However, it has been shown that vitamin D inhibits p65 nuclear translocation in a VDR-dependent manner, leading to the downregulation of NF-κB target genes [49]. Furthermore, vitamin D limits chemokine expression in adipocytes and macrophages in mouse models [50]. Interestingly, clinical studies using transcriptome data from blood patients support the immune system's involvement in the response to vitamin D3 supplementation, specifically with the downregulation of the NF-kB signaling pathway [51,52]. Therefore, further studies are necessary to show for the crosstalk between vitamin D and NF-kB activation during DENV-2 infection in human primary macrophages.

The IRF transcription factor family plays a central role in regulating immune cell activity and has consequently received growing attention. In this study, the transcription motif enrichment analysis predicted IRF5 as the TF involved in the upregulation of antiviral response genes, including *IDO1*, *ISG20*, *OASL*, *IFI44L*, *RSAD2*, *IFIT1*, *MX1*, *EPSTI1*, *CXCL10*, and *CXCL11*, in D3MDM but not in MDM. In line with this, a previous in vitro study demonstrated that VDR binds to the *IRF5* gene loci in vitamin D-treated monocytes [53]. Similarly, our analysis of publicly available ChIP-Seq and RNA-Seq datasets from vitamin D-treated macrophages, revealed that vitamin D promotes VDR binding at the proximal promoter region of *IRF5*, while VDR-KO macrophages exhibit reduced *IRF5* transcript levels. Moreover, an epigenomic study in monocytes and a transcriptomic analysis of macrophages differentiated in the presence of vitamin D showed high levels of *NOD2* [53,54]. *IRF5* is a downstream factor of the NOD2 and TLR3 signaling pathways and is essential for type I IFN signaling, thereby enhancing the antiviral state [55]. Although it has been reported that IRF5 cooperates with RelA to regulate the expression of inflammatory genes involved in macrophage polarization [56], and that vitamin D reduces lipopolysaccharide (LPS)-induced IRF5 phosphorylation, thereby attenuating the inflammatory activity of macrophages [57]. In addition, IRF5 has been shown to drive the expression of genes characteristic of M1 macrophage polarization, including TNFα, IL12, and IL-23, and play a critical role in regulating macrophage functional phenotypes [57]. Interestingly, M1 polarized MDM significantly restricts ZIKV replication, an effect associated with the upregulation of ISGs [58]. These findings suggest that IRF5 functions as a key mediator of the vitamin D-induced antiviral state in human macrophages. By promoting IRF5 expression and its transcriptional activity, vitamin D may prime macrophages to mount a robust early induction of

ISGs upon viral infection. This early ISG activation is dependent on IFN-I production and is critical for establishing an antiviral intracellular environment, thereby delaying viral replication and reducing viral antigen levels. Collectively, our findings support the notion that vitamin D orchestrates an antiviral state via the VDR-IRF5-ISG axis, contributing to the observed restriction of DENV replication.

Clinical studies performed in patients with dengue infection [59], patients with dengue and coinfected with *H. pylori* [60], and hospitalized patients with dengue fever in Pakistan [61] have shown an association between vitamin D deficiency and dengue disease progression. Furthermore, vitamin D supplementation has decreased the risk of severe dengue in patients with dengue fever [62]. Here, we show that vitamin D limits DENV-2 early replication that leads to lower induction of NF-κB and, subsequently, lower gene expression of cytokines and chemokines in primary human macrophages ([Fig 6]). Controlling the inflammatory response to DENV infection while maintaining antiviral activity may reduce disease severity and, consequently, decrease morbidity and mortality in DENV-infected patients. Vitamin D might be a prognostic factor for

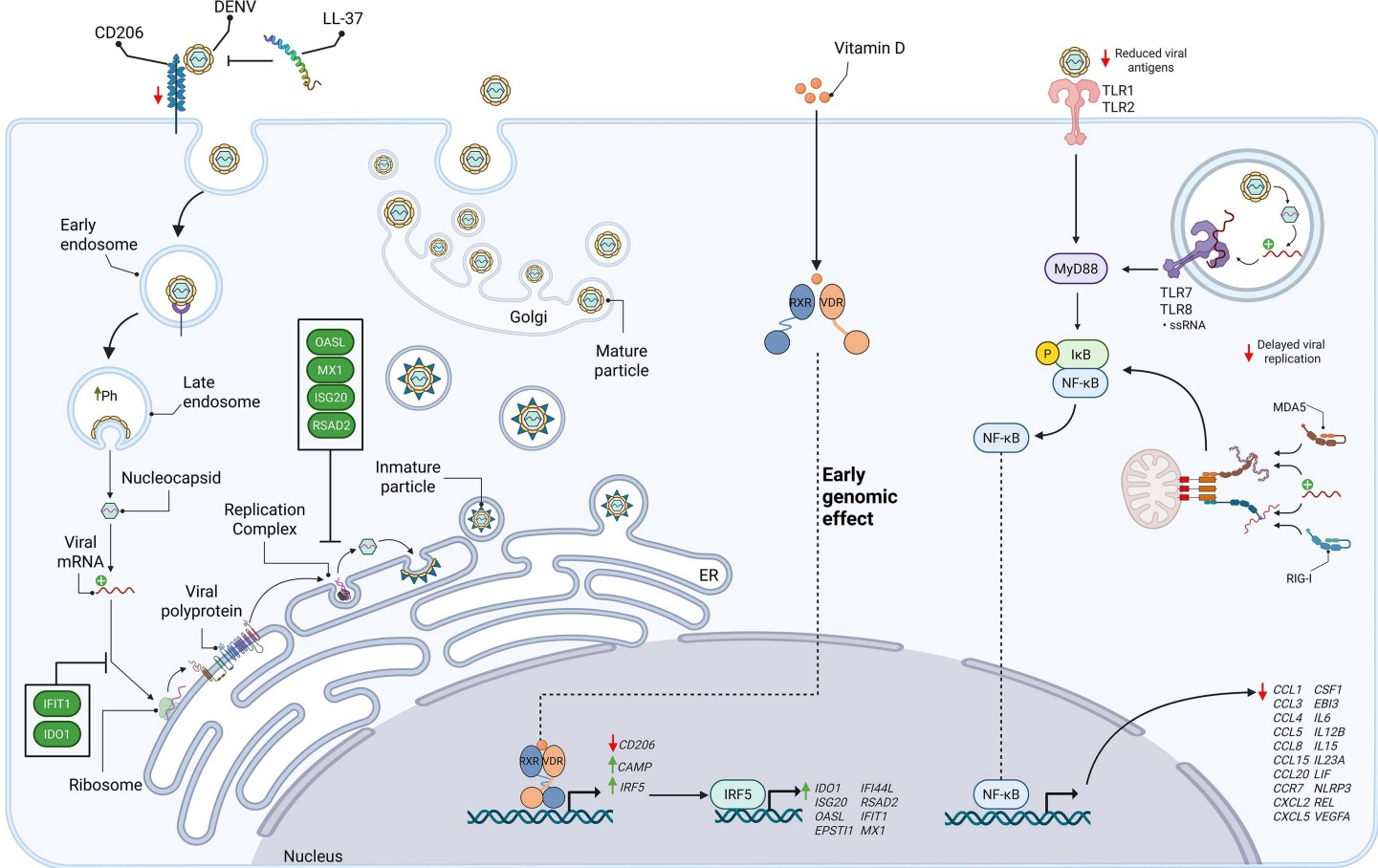

**Fig 6. Effects of vitamin D on the time course of DENV-2 infection in human primary macrophages.** Vitamin D signaling through the VDR pathway activates the transcription factor IRF5, promoting the expression of a subset of ISGs. DENV is recognized by surface or endosomal TLRs and cytoplasmic RLRs, triggering NF-κB activation and induction of the inflammatory response. Vitamin D restricts viral entry by downregulating the mannose receptor (*CD206*) and upregulating of the antimicrobial peptide LL-37 (encoded by *CAMP*). Viral particles that enter the cell release their genome, which is further restricted at the translation level (e.g., *IFIT1*, *IDO1*) and replication (e.g., *OASL*, *MX1*, *ISG20*, *RSAD2*). The reduction in viral entry and replication results in lower availability of viral antigens and replication intermediates for TLR and RLR sensing, ultimately attenuating NF-κB signaling and decreasing the expression of pro-inflammatory cytokines and chemokines. Created with BioRender.com Tamayo Molina, Y. (2025) https://BioRender.com/740eopk.

developing severe dengue disease, and its supplementation would reduce the stronger inflammatory response presented in patients infected with dengue. Vitamin D is safe, inexpensive, and widely available, and could be an effective therapeutic strategy. However, despite the findings of this study and the previously discussed literature, further research is still necessary before considering vitamin D as a therapeutic strategy for controlling dengue progression.

In summary, our previous studies demonstrated that vitamin D restricts DENV-2 replication through multiple mechanisms, including the downregulation of *CD206*, a receptor involved in virus attachment, and the upregulation of the antimicrobial peptide LL-37, both of which contribute to reduced viral entry [24]. In line with these findings, we further show that vitamin D also regulates the expression of *IRF5*, which in turn induces the expression of antiviral genes, including *IDO1*, *ISG20*, *OASL*, *IFI44L*, *RSAD2*, *IFIT1*, and *MX1*, all of which contribute to limiting viral replication. Therefore, by restricting viral entry, limiting replication, and reducing viral particle production, vitamin D indirectly attenuates activation of the NF-κB signaling pathway, resulting in decreased expression of proinflammatory cytokines and chemokines (Fig 6).

Although our kinetic gene expression analysis identifies multiple regulatory factors, this study has several limitations. First, while our data provides temporal resolution, bulk RNA sequencing limits our ability to distinguish gene expression dynamics between infected and uninfected cells. Second, the potential non-genomic effects of VDR remain to be experimentally validated. Future studies should quantify VDR localization in nuclear and cytoplasmic compartments and clarify how vitamin D modulates gene expression through NF-κB-dependent mechanisms. Another limitation is the lack of protein-level data in the present study. However, our previous work has examined protein expression during vitamin D-induced macrophage differentiation and subsequent DENV-2 infection, yielding results consistent with our transcriptomic analysis [24,25].

## Conclusion

Our findings reveal that DENV-2-infected MDM and DENV-2-infected D3MDM share a common transcriptional program following DENV-2 infection. However, vitamin D exposure modulates both the magnitude and diversity of this response, attenuating early- and mid-phase inflammatory signaling while enhancing the early antiviral activity. These results highlight a previously unrecognized role of calcitriol as a key regulator of macrophage transcriptional profile in response to DENV-2 infection.

## Supporting information

**S1 Fig. Experimental design.** Macrophages were differentiated or not in the presence of calcitriol 0.1 nM. Then, macrophages were infected with DENV-2 at an MOI of 5. Subsequently, cells were collected at 1.5-, 3-, 5.5-, 10-, and 24-h.p.i. Cells were used for bulk RNA sequencing, and supernatants were collected at 24 h.p.i. to quantify viral infectious particles. (TIF)

**S2 Fig. Piline for analysis of transcriptome data.** Transcriptome processing and analysis. Raw counts were pre-processed using batch effect correction and low-count filtering. Then, principal component analysis, differential expression analysis, and correlation analysis were performed **(A)**. Gene set enrichment using clusterProfiler and transcription factor enrichment using TFactSR were performed for differentially expressed genes in both macrophage types **(B)**. Unique genes in macrophages were used to search for biological processes using clusterProfiler and transcription factor enrichment by iRegulon. (TIF)

**S3 Fig. Analysis of all time points. Principal component analysis of all samples together (A).** Correlation analysis between samples in DENV-2 infected MDMs **(B)**. (TIF)

## Acknowledgments

The authors thank the blood bank of the "Escuela de Microbiología, UdeA, Medellín, Colombia" for providing us with leukocyte-enriched blood units from healthy individuals and the personnel at the institutions where the study was performed.

## Author contributions

**Conceptualization:** Y.S Tamayo-Molina, Juan Felipe Valdés-López, Geysson J. Fernandez, Silvio Urcuqui-Inchima.

**Data curation:** Y.S Tamayo-Molina, Juan Felipe Valdés-López, Silvio Urcuqui-Inchima.

**Formal analysis:** Y.S Tamayo-Molina, Juan Felipe Valdés-López, Geysson J. Fernandez, Silvio Urcuqui-Inchima.

**Funding acquisition:** Silvio Urcuqui-Inchima.

**Investigation:** Y.S Tamayo-Molina, Juan Felipe Valdés-López, Silvio Urcuqui-Inchima.

**Methodology:** Y.S Tamayo-Molina, Juan Felipe Valdés-López, Geysson J. Fernandez, Silvio Urcuqui-Inchima.

**Project administration:** Silvio Urcuqui-Inchima.

**Resources:** Silvio Urcuqui-Inchima.

**Software:** Y.S Tamayo-Molina, Juan Felipe Valdés-López, Geysson J. Fernandez.

**Supervision:** Silvio Urcuqui-Inchima.

**Validation:** Y.S Tamayo-Molina, Juan Felipe Valdés-López, Geysson J. Fernandez, Silvio Urcuqui-Inchima.

**Visualization:** Y.S Tamayo-Molina, Silvio Urcuqui-Inchima.

**Writing – original draft:** Y.S Tamayo-Molina, Juan Felipe Valdés-López, Geysson J. Fernandez, Silvio Urcuqui-Inchima.

**Writing – review & editing:** Y.S Tamayo-Molina, Juan Felipe Valdés-López, Geysson J. Fernandez, Silvio Urcuqui-Inchima.

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
