## [Decision Letter · Decision Letter 0]

5 Jun 2025

PONE-D-25-07206Vitamin D Enhances Antiviral Responses in Dengue Virus-Infected Macrophages by Modulating Early-Response Gene ExpressionPLOS ONE

Dear Dr. Urcuqui-Inchima,

Thank you for submitting your manuscript to PLOS ONE. After careful consideration, we feel that it has merit but does not fully meet PLOS ONE’s publication criteria as it currently stands. Therefore, we invite you to submit a revised version of the manuscript that addresses the points raised during the review process. Both reviewers raised the questions of the role of  IRF5 and the Vitamin D doses used in your experiments. Additional experiments to show the role of IRF5 and/or modification of the current statement are highly recommended before resubmission. Please carefully address all the concerns the reviewers addressed. 

We look forward to receiving your revised manuscript.

Kind regards,

Helene Minyi Liu, Ph.D.

Academic Editor

PLOS ONE

Journal Requirements:

4. We note that your Data Availability Statement is currently as follows: All relevant data are within the manuscript and in Supporting Information files.

5. Please upload a copy of Figure 7, to which you refer in your text on page 26. If the figure is no longer to be included as part of the submission please remove all reference to it within the text.

Reviewers' comments:

Reviewer's Responses to Questions

**Comments to the Author**

1. Is the manuscript technically sound, and do the data support the conclusions?

Reviewer #1: Yes

Reviewer #2: Partly

2. Has the statistical analysis been performed appropriately and rigorously? 

Reviewer #1: Yes

Reviewer #2: No

3. Have the authors made all data underlying the findings in their manuscript fully available?

Reviewer #1: Yes

Reviewer #2: Yes

4. Is the manuscript presented in an intelligible fashion and written in standard English?

Reviewer #1: Yes

Reviewer #2: Yes

5. Review Comments to the Author

Reviewer #1: In this manuscript, Tamayo-Molina et. al. explore the impact of vitamin D on antiviral responses in dengue virus (DENV)-infected human monocyte-derived macrophages (MDMs). The authors apply a kinetic transcriptomic approach to characterize early, mid, and late gene expression patterns during infection, highlighting key transcriptional differences modulated by vitamin D treatment. This work addresses a timely and clinically relevant question, considering the lack of specific antiviral therapies for dengue fever. I have several concerns as follows:

1. Non-genomic vs Genomic Effects: While the authors suggest non-genomic mechanisms of vitamin D at physiological concentrations, further evidence is needed to support this claim. As acknowledged in the limitations, additional validation such as VDR localization studies or functional assays distinguishing genomic from non-genomic actions would greatly strengthen the manuscript.

2. Infection Heterogeneity: Given that bulk RNA-seq was performed, the heterogeneity of infected versus bystander cells could confound interpretations. A brief discussion or data (e.g., infection rates, percentage of infected cells) could help contextualize transcriptomic changes.

3. Vitamin D Dose Justification: The distinction between the effects of 0.1 nM and 1 nM vitamin D is intriguing. However, it would benefit the reader if the authors could justify the clinical relevance of the chosen concentration (0.1 nM) and its pharmacological or physiological basis.

Minor comments:

• In the Introduction, the background on IRF5 and its known antiviral roles could be expanded.

• A more explicit statement on how this work advances previous studies by the same group would clarify the novelty.

• Please ensure consistent gene naming (e.g., VDR, NF-κB subunits) throughout the text.

• Consider editing for minor grammatical improvements and flow in the Discussion section.

Reviewer #2: The legend of Figure 1A indicates that the concentration of VitD used was 0.1 nM. However, in Figures 1C and 1D, it is suggested that 0.1 nM is insufficient to exert an effect. This inconsistency should be clarified, and the rationale for the concentration used in each panel needs to be justified.

In Figure 2, there is a spelling error that should be corrected. Regarding Figure 2A, the data showed that the concentration of VitD used may have been insufficient to suppress viral replication. It is recommended that the authors test a broader or higher range of concentrations to fully evaluate the antiviral potential of vitamin D. Additionally, in Figure 2D, the numerical data shown in the figure do not align with the values described in the main text.

In Figure 5, several issues affect the clarity and interpretation of the data. First, there are some instances of mislabeling or mismatched labels between the figure panels and the corresponding text. Second, in Figure 5H, the data do not clearly support the conclusion stated in the text. The term “early stage of infection” is mentioned in the text but is not clearly defined in terms of specific time points, making it difficult to assess the relevance of the findings. The authors should specify the exact time frame referred to as the early stage and show corresponding data that support their claims.

Although the manuscript proposed that vitamin D modulates viral infection through the VDR signaling pathway, the current data do not provide strong support for this mechanism. The changes observed in the VDR pathway after vitamin D treatment are minimal, and it remains unclear which downstream target proteins are significantly affected. Without clear identification of these targets, it is difficult to draw conclusions about how vitamin D might influence viral replication. Furthermore, the manuscript suggested potential involvement of NF-κB and IRF5, but no direct evidence is presented to establish a mechanistic link between vitamin D signaling and these immune regulators. Additional experiments are needed to substantiate these claims and clarify the molecular pathways involved.

6. PLOS authors have the option to publish the peer review history of their article (what does this mean? ). If published, this will include your full peer review and any attached files.

**Do you want your identity to be public for this peer review?** For information about this choice, including consent withdrawal, please see our Privacy Policy .

Reviewer #1: No

Reviewer #2: No

---

## [Author Response · Author response to Decision Letter 1]

2 Jul 2025

Thank you for your email and the positive feedback on our paper “Vitamin D Enhances Antiviral Responses in Dengue Virus-Infected Macrophages by Modulating Early-Response Gene Expression”. PONE-D-25-07206. We sincerely appreciate the very constructive comments of the reviewer.

Reviewer #1:

In this manuscript, Tamayo-Molina et. al. explore the impact of vitamin D on antiviral responses in dengue virus (DENV)-infected human monocyte-derived macrophages (MDMs). The authors apply a kinetic transcriptomic approach to characterize early, mid, and late gene expression patterns during infection, highlighting key transcriptional differences modulated by vitamin D treatment. This work addresses a timely and clinically relevant question, considering the lack of specific antiviral therapies for dengue fever. I have several concerns as follows:

Majos comments:

1. Non-genomic vs Genomic Effects: While the authors suggest non-genomic mechanisms of vitamin D at physiological concentrations, further evidence is needed to support this claim. As acknowledged in the limitations, additional validation such as VDR localization studies or functional assays distinguishing genomic from non-genomic actions would greatly strengthen the manuscript.

We thank the referee for the observation. Indeed, we agree with your point of view. However, we cannot access such validations involving VDR localization or functional assays. Because of that, we focus the manuscript on the genomic effects of vitamin D, supporting the role of IRF5-mediated antiviral responses. We added in the limitations of the study this concern.

In the discussion, in line 463, we added the following:

“However, we could not assess these effects in the present study.”

In the limitations, we added in lines 532 to 535 the following:

“Second, the potential non-genomic effects of VDR remain to be experimentally validated. Future studies should quantify VDR localization in nuclear and cytoplasmic compartments and clarify how vitamin D modulates gene expression through NF-κB-dependent mechanisms.”

2. Infection Heterogeneity: Given that bulk RNA-seq was performed, the heterogeneity of infected versus bystander cells could confound interpretations. A brief discussion or data (e.g., infection rates, percentage of infected cells) could help contextualize transcriptomic changes.

We added the following lines to attend to the referee’s observation in the discussion section. Lines 400 to 444:

“As was observed in previous studies conducted in our laboratory [24, 25], we found that DENV-2-infected D3MDM exhibited a lower percentage of infected cells (~25%) compared to conventional MDMs (~15%) at MOI of 5. This reduction infection was associated with decreased mRNA expression of TLR3 and TLR7, along with increased expression of antiviral genes, including PKR and OAS1 [22–25].”

3. Vitamin D Dose Justification: The distinction between the effects of 0.1 nM and 1 nM vitamin D is intriguing. However, it would benefit the reader if the authors could justify the clinical relevance of the chosen concentration (0.1 nM) and its pharmacological or physiological basis.

We thank the referee for this suggestion. We added in lines 217 to 222 the following:

The concentration of vitamin D used in this study (0.1 nM) was selected to mimic physiological levels of circulating 1,25(OH)₂D₃, which typically range between 0.04 and 0.150 nM in serum. This concentration also corresponds to the high-affinity dissociation constant (Kd = ~0.1 nM) of 1,25(OH)₂D₃ for the VDR [35]. Furthermore, we previously reported that macrophages differentiated in the presence of 0.1 nM vitamin D (D3MDM) exhibited reduced susceptibility to DENV infection [24, 25].

Minor comments:

• In the Introduction, the background on IRF5 and its known antiviral roles could be expanded.

It was done

• A more explicit statement on how this work advances previous studies by the same group would clarify the novelty.

It was done

• Please ensure consistent gene naming (e.g., VDR, NF-κB subunits) throughout the text.

It was done

• Consider editing for minor grammatical improvements and flow in the Discussion section.

It was done

Reviewer #2:

1. The legend of Figure 1A indicates that the concentration of VitD used was 0.1 nM. However, in Figures 1C and 1D, it is suggested that 0.1 nM is insufficient to exert an effect. This inconsistency should be clarified, and the rationale for the concentration used in each panel needs to be justified.

We agree with the reviewer comments and have modified Figures 1C and 1D accordingly. The updated figures provide a more precise representation of the transcriptional changes induced by Vitamin D. These modifications led to slight but noteworthy changes, which have been incorporated into the main text in lines 232 to 239, as follows:

“The DEGs analysis revealed 16 upregulated and 43 downregulated genes in D3[0.1 nM]MDM, while 100 genes were upregulated and 56 downregulated in D3[1 nM]MDM (Fig 1C). Gene overrepresentation analysis using WikiPathways showed that both concentrations induced the expression of genes associated with the vitamin D receptor pathway (WP2877), including CLMN, G0S2, CD14, TREM1, and CAMP, which were regulated in both concentrations (Fig 1D). In contrast, the higher vitamin D concentration also induced the expression of additional genes, such as CYP24A1, MYC, ALOX5, HILPDA, and S100A8 (Fig 1E). .”

2. In Figure 2, there is a spelling error that should be corrected. Regarding Figure 2A, the data showed that the concentration of VitD used may have been insufficient to suppress viral replication. It is recommended that the authors test a broader or higher range of concentrations to fully evaluate the antiviral potential of vitamin D. Additionally, in Figure 2D, the numerical data shown in the figure do not align with the values described in the main text.

We thank the reviewer for this suggestion. We have added the antiviral activity of macrophages differentiated with 1 nM of Vitamin D (Figures 1A and 1B). With these changes, we explore a broader dose of vitamin D. However, we cannot access genome replication as in Figure 2A for Vitamin D 1 nM. In this context, we added in lines 222 to 227 the following phases:

“As shown in Figure 1A, D3MDM were consistently less susceptible to DENV-2 infection than MDM. The percentage of inhibition reached approximately 35% and 45% in D3MDM differentiated with 0.1 nM (D3[0.1 nM]MDM) and 1 nM (D3[1 nM]MDM) of vitamin D, respectively, compared to MDM at 24 h.p.i. (Fig 1B). It is worth noting that the antiviral effect of vitamin D appears to be dose dependent.”

Regarding Figure 2D, we apologize for this mistake; we have corrected this inconsistency between the number of upregulated genes in the text and figure.

3. In Figure 5, several issues affect the clarity and interpretation of the data:

a. First, there are some instances of mislabeling or mismatched labels between the figure panels and the corresponding text.

b. Second, in Figure 5H, the data do not clearly support the conclusion stated in the text. The term “early stage of infection” is mentioned in the text but is not clearly defined in terms of specific time points, making it difficult to assess the relevance of the findings. The authors should specify the exact time frame referred to as the early stage and show corresponding data that support their claims.

We apologize for this mistake. We have corrected the mislabeling in Figure 5. In addition, we defined the specific time points regarding early, mid, and late effects. Therefore, in lines 250 to 254, we added which stage corresponds to these time points:

“The early time points (1.5-, 3-, and 5.5- h.p.i), and the mid time point (10 h.p.i). were selected to capture kinetic changes in the transcriptional profile associated with early viral entry and replication, independent of the effects of infectious virus release. The 24 h.p.i. time point was chosen as the late time of infection, corresponding to the assembly and release of new viral progeny.”

Importantly, we introduced through the text the analysis of the results in line with these three time points: early, mid and late.

4. Although the manuscript proposed that vitamin D modulates viral infection through the VDR signaling pathway, the current data do not provide strong support for this mechanism. The changes observed in the VDR pathway after vitamin D treatment are minimal, and it remains unclear which downstream target proteins are significantly affected. Without clear identification of these targets, it is difficult to draw conclusions about how vitamin D might influence viral replication. Furthermore, the manuscript suggested potential involvement of NF-κB and IRF5, but no direct evidence is presented to establish a mechanistic link between vitamin D signaling and these immune regulators. Additional experiments are needed to substantiate these claims and clarify the molecular pathways involved.

We thank the referee for this important suggestion. We have modified Figure 4B and added available data in GEO regarding the mechanistic link between vitamin D signaling and these immune regulators in human macrophages. Therefore, we added two new figures: Figure 4C and 4D. We hope this helps us understand the biological significance of Vitamin D IRF5-mediated antiviral response. In the text, in lines 370 to 378, we added the following:

“To determine whether vitamin D directly regulates IRF5 and NFKB1 gene expression, we reanalyzed VDR ChIP-Seq data (GSE89431) and transcriptomic data from wild-type (TW) and VDR-knockout (KO) THP-1 cells (GSE157514), at both early (2 and 4 h) and late (24 h) time points. VDR binding was consistently observed at the proximal promoter of IRF5 from early to late time points, whereas no VDR occupancy was detected at the NFKB1 locus (Fig 4C). Accordingly, IRF5 transcript levels (TPM) increased significantly in vitamin D-treated WT THP-1 cell (D3-VDR-THP1), but not in vitamin D-treated VDR-KO THP-1 cells (D3-KO-THP1), where IRF5 expression remained comparable to that of untreated controls (Fig 4D). .”

In addition, we cannot provide further evidence of the mechanistic link between NF-κB and the vitamin D pathway. However, to further expand the effect of macrophages differentiated with vitamin D on gene expression, we have conceived a new subfigure in Figure 3. We added the following in lines 345 to 352:

“In addition, we observed temporal changes in the expression of these transcription factors during infection. In both macrophage types, VDR mRNA levels declined from 0 to 3 h.p.i., peaked at 5.5 h.p.i., and then decreased again from 10 to 24 h.p.i. (Fig 3D) Notably, VDR expression was consistently higher in D3MDMs than in MDMs between 0 and 3 h.p.i. We also found that JUN, RELA, and NFKB1 mRNA expression peaked at 3 h.p.i., declined at 10 h.p.i., and increased again at 24 h.p.i. Notably, at their peak expression, the levels of these TFs were lower in D3MDMs than in MDMs, with the most pronounced difference observed for NFKB1 (Fig 3D).”

---

## [Decision Letter · Decision Letter 1]

29 Jul 2025

PONE-D-25-07206R1Vitamin D Enhances Antiviral Responses in Dengue Virus-Infected Macrophages by Modulating Early-Response Gene ExpressionPLOS ONE

Dear Dr. Urcuqui-Inchima,

Thank you for submitting your manuscript to PLOS ONE. After careful consideration, we feel that it has merit but does not fully meet PLOS ONE’s publication criteria as it currently stands. Therefore, we invite you to submit a revised version of the manuscript that addresses the points raised during the review process.

No further experiments are required. However, statistic analysis of Figure 3 and Discussion of  the functional link between IRF5 and vitamin D signaling is required.

We look forward to receiving your revised manuscript.

Kind regards,

Helene Minyi Liu, Ph.D.

Academic Editor

PLOS ONE

Journal Requirements:

Additional Editor Comments:

No further experiments are required. However, statistic analysis of Figure 3 and Discussion of the functional link between IRF5 and vitamin D signaling is required.

Reviewers' comments:

Reviewer's Responses to Questions

**Comments to the Author**

1. If the authors have adequately addressed your comments raised in a previous round of review and you feel that this manuscript is now acceptable for publication, you may indicate that here to bypass the “Comments to the Author” section, enter your conflict of interest statement in the “Confidential to Editor” section, and submit your "Accept" recommendation.

Reviewer #1: All comments have been addressed

Reviewer #2: (No Response)

2. Is the manuscript technically sound, and do the data support the conclusions?

Reviewer #1: Partly

Reviewer #2: Partly

3. Has the statistical analysis been performed appropriately and rigorously? 

Reviewer #1: Yes

Reviewer #2: Yes

4. Have the authors made all data underlying the findings in their manuscript fully available?

Reviewer #1: Yes

Reviewer #2: Yes

5. Is the manuscript presented in an intelligible fashion and written in standard English?

Reviewer #1: Yes

Reviewer #2: Yes

6. Review Comments to the Author

Reviewer #1: (No Response)

Reviewer #2: The authors have made meaningful revisions in response to previous comments, and the manuscript has improved. However, several concerns remain that need to be addressed to strengthen the conclusions:

1. In Figures 3C and 3D, statistical analysis is not indicated. Without clear markers of significance, it is difficult to evaluate the reliability and relevance of the observed differences.

2. The study relies primarily on RNA-seq data and lacks molecular validation. Additional experiments, such as western blotting to assess IRF5-related proteins and viral proteins levels, are needed to support the claim that vitamin D has antiviral activity and could reduce inflammation.

3. The manuscript reports an increase in IRF5 expression following vitamin D treatment combined with viral infection. However, the functional link between IRF5 and vitamin D signaling remains unclear. The authors should further explain the relevance of this finding and how it contributes to the proposed antiviral mechanism.

7. PLOS authors have the option to publish the peer review history of their article (what does this mean? ). If published, this will include your full peer review and any attached files.

**Do you want your identity to be public for this peer review?** For information about this choice, including consent withdrawal, please see our Privacy Policy .

Reviewer #1: No

Reviewer #2: No

---

## [Author Response · Author response to Decision Letter 2]

4 Aug 2025

Point-by-point response to reviewers

Thank you for your email and the positive feedback on our paper “Vitamin D Enhances Antiviral Responses in Dengue Virus-Infected Macrophages by Modulating Early-Response Gene Expression”. PONE-D-25-07206.

Reviewer #2: The authors have made meaningful revisions in response to previous comments, and the manuscript has improved. However, several concerns remain that need to be addressed to strengthen the conclusions:

1. In Figures 3C and 3D, statistical analysis is not indicated. Without clear markers of significance, it is difficult to evaluate the reliability and relevance of the observed differences.

Answer: We thank the reviewer for this observation. We have now included statistical analysis for Figure 3D using a two-way ANOVA model (Expression ~ Treatment * Time), followed by post hoc unpaired t-tests at each time point. However, statistical analysis could not be performed for Figure 3C, as there is only a single value of regulated genes (K) per condition at each time point. Thus, no biological replicates are available for comparison.

2. The study relies primarily on RNA-seq data and lacks molecular validation. Additional experiments, such as western blotting to assess IRF5-related proteins and viral proteins levels, are needed to support the claim that vitamin D has antiviral activity and could reduce inflammation.

Answer: We thank the reviewer for their insightful comment and suggestions. However, we would like to clarify that the primary objective of this study was to perform a transcriptomic analysis of MDMs differentiated in the presence or absence of vitamin D and subsequently infected with DENV. We aimed to gain a deeper molecular understanding of the functional and antiviral responses modulated by vitamin D in this context.

3. The manuscript reports an increase in IRF5 expression following vitamin D treatment combined with viral infection. However, the functional link between IRF5 and vitamin D signaling remains unclear. The authors should further explain the relevance of this finding and how it contributes to the proposed antiviral mechanism.

We have added the following lines to address the reviewer’s comment in the Discussion section (lines 499–513):

“These findings suggest that IRF5 functions as a key mediator of the vitamin D-induced antiviral state in human macrophages. By promoting IRF5 expression and its transcriptional activity, vitamin D may prime macrophages to mount a robust early induction of ISGs upon viral infection. This early ISG activation is dependent on IFN-I production and is critical for establishing an antiviral intracellular environment, thereby delaying viral replication and reducing viral antigen levels. Collectively, our findings support the notion that vitamin D orchestrates an antiviral state via the VDR-IRF5-ISG axis, contributing to the observed restriction of DENV replication. Notably, IRF5 has been shown to drive the expression of genes characteristic of M1 macrophage polarization, including TNFα, IL12, and IL-23, and play a critical role in regulating macrophage functional phenotypes [57]. Interestingly, M1 polarized MDM significantly restricts ZIKV replication, an effect associated with the upregulation of ISGs [58].

---

## [Editor Report · Decision Letter 2]

6 Aug 2025

Vitamin D Enhances Antiviral Responses in Dengue Virus-Infected Macrophages by Modulating Early-Response Gene Expression

PONE-D-25-07206R2

Dear Dr. Urcuqui-Inchima,

We’re pleased to inform you that your manuscript has been judged scientifically suitable for publication and will be formally accepted for publication once it meets all outstanding technical requirements.

Kind regards,

Helene Minyi Liu, Ph.D.

Academic Editor

PLOS ONE
---

## [Editor Report · Acceptance letter]

PONE-D-25-07206R2

PLOS ONE

Dear Dr. Urcuqui-Inchima,

I'm pleased to inform you that your manuscript has been deemed suitable for publication in PLOS ONE. Congratulations! Your manuscript is now being handed over to our production team.

Kind regards,

on behalf of

Dr. Helene Minyi Liu

Academic Editor

PLOS ONE